# Sublinear transport in Kagome metals from the interplay of Dirac cones and Van Hove singularities

Nikolai Peshcherenko[1], Ning Mao[1], Claudia Felser [1] & Yang Zhang [2,3] ✉

Recent experiments have revealed that many kagome metals exhibit sublinear temperature dependence in resistivity up to room temperature. Here, we develop a minimal semiclassical two-pocket model—comprising a Dirac cone and a Van Hove singularity—and show that, within an extended Fermi liquid scattering framework, internode electron-electron interactions naturally lead to sublinear scaling in both electrical and thermal transport at low temperatures. At higher temperatures, distinct scattering channels for charge and heat currents lead to a violation of the Wiedemann-Franz law. Our work provides a simple and broadly applicable framework for understanding anomalous transport in kagome metals, capturing non-Fermi-liquid behavior without requiring fine-tuning or exotic interactions.

In itinerant metals, electrons/holes are the dominant charge carriers, and the Fermi liquid framework typically applies. Consequently, a clean system is expected to exhibit a resistivity scaling with $T^2$ at intermediate temperatures, reflecting electron-electron interactions and signaling Fermi liquid behavior. The deviations from this well-known result are studied both experimentally and theoretically as the signatures of strongly interacting systems, including Luttinger liquid systems[1], heavy fermion metals[2], underdoped cuprates[3], graphene at charge neutrality point[4,5], etc. These deviations, in terms of resistivity scaling, could be both superlinear (meaning resistivity scaling with temperature $T$ as $\rho(T) \propto T^n$, $n > 1$, $n \neq 2$) and linear. The so-called strange metal state featuring $T$ linear scaling was previously observed in refs. 6–10 and superlinear temperature behavior was predicted for materials in the vicinity of quantum phase transition[11,12]. Although these systems have quite different properties, the non-Fermi liquid behaviors are all linked with emerging collective excitations due to strong interactions.

Recently, Kagome lattices have emerged as pivotal platforms for exploring strongly correlated phenomenon[13,14], topological magnetism[15–21], and unconventional superconductivity[22]. Experiment on Kagome metal Ni$_3$In[23] has demonstrated a quite unexpected and well-pronounced sublinear in $T$ resistivity ($\rho(T) \propto T^n$, $n < 1$) at relatively high temperatures $T \gtrsim 100$ K, where strong quantum fluctuations or

collective excitations are unlikely to appear. Similar sublinear behavior, albeit less pronounced, has been observed in three-dimensional Kagome compounds like ScV$_6$Sn$_6$, CsV$_3$Sb$_5$, RbV$_3$Sb$_5$, KV$_3$Sb$_5$[24–30] and Yb$_{0.5}$Co$_3$Ge$_3$[31,32], MgCo$_6$Ge$_6$[33] around room temperature (see also Table 1). Therefore, the most plausible explanation for this type of behavior should rather be semiclassical. However, existing semiclassical scattering contributions from electron-phonon coupling predict $\rho(T) \propto T^n$, $n \geq 1$[34,35], and impurity scattering is known to give $T$-independent contribution.

In this work, we introduce a two-pocket model consisting of Dirac cone and Van Hove singularity, which allows a sublinear in temperature scaling of resistivity as observed in the family of Kagome materials. Our model study focuses exclusively on single electron excitations, which are semiclassically described using the Boltzmann equation. The crucial ingredient is the internode electron-electron interaction, which is enhanced due to the divergent density of states from VHS at Fermi level. We demonstrate that even though this internode scattering conserves momentum, its presence alongside momentum-relaxing processes (like impurity scattering) generates a leading $T$-sublinear contribution to resistivity, with a different sublinear exponent also emerging for thermal conductivity. Furthermore, we show that this model predicts a breakdown of the Wiedemann-Franz law and a non-trivial temperature dependence of the Lorentz number. At

[1]Max Planck Institute for Chemical Physics of Solids, Dresden, Germany. [2]Department of Physics and Astronomy, University of Tennessee, Knoxville, Tennessee, USA. [3]Min H. Kao Department of Electrical Engineering and Computer Science, University of Tennessee, Knoxville, Tennessee, USA. ✉e-mail: yangzhang@utk.edu

**Table 1 | Temperature scaling exponent γ (from experimental data) and Van Hove singularity position with respect to the Fermi level (taken from DFT calculations) for certain materials from the kagome family**

| Material | Exponent γ, $\rho(T) \propto T^\gamma$ | Temperature range, K | VHS position, meV | ARPES VHS signature | Effective mass, $m_e$ |
|---|---|---|---|---|---|
| $Ni_3In$ | 0.3[23] | 100–300 | 10[23], Fig. 4 | 23 | – |
| $ScV_6Sn_6$ | 0.6[24] | 91–400 | 80[24] | 58 | 0.5[59] |
| $CsV_3Sb_5$ | 0.6[24,25] | 94–300 | 30[60] | 61,62 | 0.03[63] |
| $RbV_3Sb_5$ | 0.6[24,28] | 102–300 | 70[60] | 42,62 | 0.1[28] |
| $KV_3Sb_5$ | 0.6[24,26] | 78–300 | 60[60] | 64 | 0.1[65] |
| $Yb_{0.5}Co_3Ge_3$ | 0.6[31,32] | 100–300 | 20[66] | – | – |
| $MgCo_6Ge_6$ | 0.8[33] | 100–300 | 50[66] | – | – |
| $Ni_3Sn$ | 1[23] | 50–300 | 100[23] | 23 | – |

The temperature range exhibiting nonlinear behavior is listed, with the upper temperature limit corresponding simply to the room temperature. The lower temperature limit for V-based materials corresponds to the charge density wave transition temperature that would bring[24] the Fermi level closer to the VHS. For the VHS being sufficiently close to the Fermi level, γ < 1 due to enhanced electron-electron scattering. For a remote VHS, γ = 1 due to electron-phonon scattering[35]. Experimental evidence for the Van Hove singularity and for Dirac pocket (in the form of a small effective mass obtained from Shubnikov-de Haas measurement) is listed.

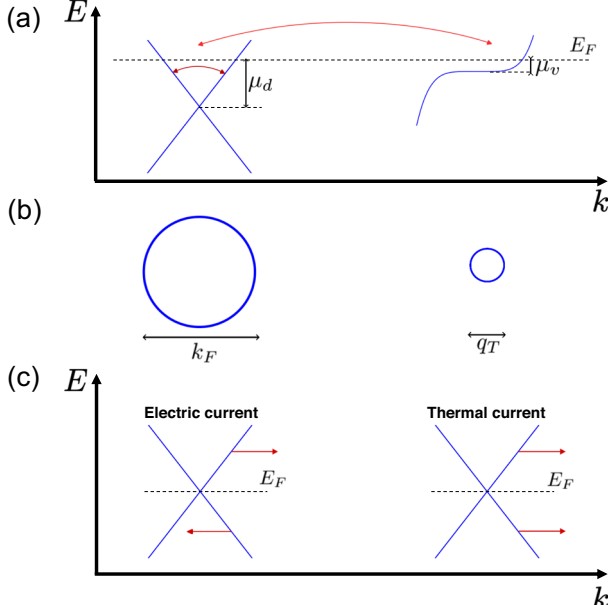

**Fig. 1 | Schematics for a kagome metal band structure. a** Two-pocket model of a Kagome metal. Node $d$ is a Dirac pocket with a relatively large Fermi surface. Fast Dirac electrons relax due to both intranode (shown as dark red arrow) and internode (bright red) electron-electron scattering. Node $v$ has weak dispersion $E_v \sim Ak^\alpha$, $\alpha > 2$ and $\mu_d, \mu_v$ are local chemical potential. **b** Thermally activated Fermi pockets for $\mu_v \ll T \ll \mu_d$. For $T \gg \mu_v$ the effective size of Fermi surface $q_T$ (which is the size of the thermally activated electrons cloud in the momentum space close to the VHS) at node $v$ is $T$-dependent: $Aq_T^\alpha \lesssim T$, or, equivalently, $q_T \lesssim (T/A)^{1/\alpha}$. Due to the weak dispersion of VHS electrons, fast Dirac electrons scattering process is almost energy-conserving. This internode process is dominant due to high $v_v$ and allows for sublinear $T$ dependence. **c** For $T \gg \mu_d$, intranode scattering is able to relax only charge but not thermal current. Here, an electron-hole pair taken at the same momentum provides opposite in sign contributions to charge transport, but the same sign for heat transport. Thus, contribution to heat transport is proportional to total momentum and could not be relaxed by intranode scattering[52].

temperatures higher than Dirac node chemical potential, both electrons and holes are thermally activated near the Dirac node. And the intranode scattering in Dirac systems is able to relax electric but not thermal current. Thus, thermal current is relaxed only with internode scattering. In our setup, the dominant electric and thermal carriers are essentially the same, but they are relaxed by intranode and internode scattering, respectively.

## Results
### Two-pocket model
The abundant exotic states in Kagome materials are deeply linked to their remarkable band structures. These band structures consist of a Dirac node, a saddle point Van Hove singularity (VHS) and a flat band[36–38], which was proved by both density functional theory calculations[26,39–41] as well as by experimental evidence such as ARPES[25,42] and scanning tunneling microscope spectroscopy. For the relevant material examples, also see Table 1.

We start with an explicit introduction of a two-pocket model (see Fig. 1), as flat band is generally away from Fermi level. The fast pocket is basically a Dirac pocket with an isotropic Dirac spectrum

$$E_d(\mathbf{k}) = v_F|\mathbf{k}| \tag{1}$$

and the slow pocket has saddle point type of momentum dispersion in $k_z = 0$ plane:

$$E_v(\mathbf{k}) = A\left(|k_x|^\alpha - |k_y|^\alpha\right). \tag{2}$$

We denote two pockets as $d$ (Dirac) and $v$ (VHS), correspondingly. It is important to note that in our model, the $v$ pocket refers only to thermally activated electrons closest to the VHS, with the exact size $q_T$ of the relevant electronic cloud to be defined below. Although the full Fermi pocket near the VHS can be much larger, only electrons in the immediate vicinity of the VHS are relevant for the scattering channel considered in this work.

Density of electronic states from node $v \nu_v(E)$ demonstrates VHS behavior:

$$\nu_v(E) = \sum_{\mathbf{k}} \delta\left(E - E_v(\mathbf{k})\right) \propto \frac{1}{\alpha A}\left(\frac{A}{|E|}\right)^\beta,$$
$$\beta = 1 - \frac{2}{\alpha}. \tag{3}$$

for $\alpha > 2$. Since having a singular density of states is crucial for our analysis, in what follows, we consider exclusively $\alpha > 2$. This saddle point VHS is located near $M$ momentum in Kagome metals such as $Ni_3In$ and $ScV_6Sn_6$ (see discussion in "Experimental data comparison" section and in Supplementary Information, section II). Since $v$ electrons near the VHS are slow (have relatively small Fermi velocities), all transport currents are mainly carried by fast Dirac electrons $d$.

However, $v$ pocket electrons still influence transport properties. This is due to their high density of states, which enhances the

(a)

(b)

**Fig. 2 | Intra- and internode scattering rates for fast current-carrying Dirac electrons.** Electron-electron scattering rate as a function of temperature for **a** Dirac intranode scattering and **b** internode scattering processes. For $T \gg \mu_v$, the effective size of $v$ pocket is $T$-dependent, which gives rise to sublinear $T$ dependence. For $T \ll \mu_d$ internode scattering is dominant due to high density of states in $v$ pocket. However, for $T \gg \mu_d$ intranode scattering is stronger since intranode interactions are not screened. Coupling constant $\alpha = e^2/\hbar v_F$ is assumed to be small.

scattering rate of internode electron-electron interactions. These interactions in the experimentally relevant limit $T \gg \mu_v$[23,24] could manifest themselves in sublinear temperature scaling of kinetic coefficients, with $\mu_v$ being defined at Fig. 1a. Namely, for interpocket electron-electron scattering, the electrons closest to the Van Hove singularity provide the leading contribution (due to singular density of states) to the scattering rate. In the limit $T \gg \mu_v$, the size of the thermally activated electrons cloud near VHS in momentum space is $q_T \propto T^{1/\alpha}$ (see description for Fig. 1). At the same time, $E_v(\mathbf{q})$ demonstrates relatively weak dispersion in the vicinity of the Van Hove singularity. Thus, the scattering of Dirac electrons at $v$ pocket electrons proceeds with negligible energy transfer. This is why $v$ electrons could be treated as a momentum-relaxing reservoir with its size $q_T$ dependent on $T$. We show below that the observed sublinear $\rho(T)$ behavior[23–28] stems from this $T$-dependent effective $v$ pocket size $q_T$.

We begin with a detailed study of the relatively large Dirac Fermi surface limit for $T \ll \mu_d$, so that a standard low-$T$ quasiparticle picture holds. Nevertheless, we also give consideration to $T \gg \mu_d$ case as it is experimentally accessible and, as we will show below, reveals additional observable physical phenomena.

## Boltzmann equation description
Fast current-carrying Dirac electrons distribution function $f_d$ obeys semiclassical Boltzmann equation:

$$\partial_t f_d + \mathbf{v} \cdot \partial_\mathbf{r} f_d + e\mathbf{E} \cdot \partial_\mathbf{p} f_d = I_{\text{intra}} + I_{\text{inter}} + \\ + I_{\text{imp}} + I_{\text{nc}}, \quad (4)$$

where $I_{\text{intra}}$, $I_{\text{inter}}$ stand for intra- and internode electron-electron scattering, $I_{\text{imp}}$ is for impurity scattering, and $I_{\text{nc}}$ describes other types of momentum-relaxing contributions (e.g., scattering at phonons or Umklapp processes for electron-electron interactions).

Below Debye temperature in Kagome metals[23–28] e-e interaction appears to play a major role in transport phenomena. As we show in what follows, this is due to the sublinear scaling of scattering rate with temperature being contributed to exclusively by e-e scattering. Other types of contributions, such as electron-phonon or impurity scattering, provide only superlinear $T$ dependence[34] for electron-phonon exchange and $T$ independent contribution for impurity scattering. Since in metallic systems, electron-electron interaction is screened, we model it in the form of contact interaction with amplitude $g$:

$$H_{\text{int}}(\mathbf{r}, \mathbf{r}') = g\delta(\mathbf{r} - \mathbf{r}'), \quad (5)$$

with an exception for $T \gg \mu_d$ case (see below). This assumption is justified for typical momentum transfers that are small compared to the inverse screening length. Umklapp processes for electron-electron scattering are very rare due to the very large momentum transfer required and are therefore omitted.

Electron-electron scattering processes could be basically of two types: intranode (happening between electrons from the same node) or internode (between different nodes), for an outline see Fig. 2. We tackle these two processes separately.

## Intranode scattering
We first note that intranode processes within $v$ pocket are not relevant to transport properties, since the corresponding electronic states possess low group velocity and thus provide only a small direct contribution to transport currents. Therefore, we focus on Dirac electrons scattering. The corresponding scattering rate was previously evaluated in refs. [43–45]. Result for electron-electron relaxation time depends on the relation between Dirac fermions chemical potential $\mu_d$ (see Fig. 1a) and temperature $T$. For relaxation rate $\tau_{\text{intra}}^{-1}$ we have:

$$\tau_{\text{intra}}^{-1} \propto \left(\frac{e^2}{\hbar v_F}\right)^2 \begin{cases} \frac{T^2}{\mu_d}, & T \ll \mu_d, \\ T, & T \gg \mu_d, \end{cases} \quad (6)$$

so that a crossover between Fermi ($\propto T^2$) and high temperature behavior happens at $T \sim \mu_d$.

## Internode scattering
Internode electron-electron scattering appears to be a crucial ingredient for sublinear $T$ behavior of transport coefficients. As we show below, this behavior is due to the weakly dispersing VHS band. Namely, due to the weak dispersion of these states, internode scattering of fast current-carrying Dirac electrons proceeds with negligible energy transfer. Thus, the expression for the internode transport scattering rate takes the following simple form:

$$\frac{1}{\tau_{\text{e} - \text{e}}} \sim n_v g^2 \sum_\mathbf{q} \delta(E(\mathbf{p}) - E(\mathbf{p} + \mathbf{q}))(1 - \cos \theta_{\mathbf{p}+\mathbf{q},\mathbf{p}}), \quad (7)$$

where $\theta_{\mathbf{p}+\mathbf{q},\mathbf{p}}$ describes Dirac electron's scattering angle between its initial $\mathbf{p}$ and final $\mathbf{p} + \mathbf{q}$ momenta, $n_v$ is the concentration of VHS electrons. The phase volume of scattered electrons in Eq. (7) is, however, limited by the magnitude of transferred momentum $\mathbf{q}$. The upper limit for $\mathbf{q}$ could be, in turn, estimated by the size of momentum-relaxing reservoir of VHS states $q_T \lesssim (T/A)^{1/\alpha}$ (see Fig. 1b). It is important to note that even for an anisotropic Van Hove singularity, the estimate for the transmitted momentum $q_T \propto T^{1/\alpha}$ would hold true, allowing for sublinear resistivity scaling discussed below. Assuming small momentum transfer $q_T \ll p_F$ (which is true for $T \ll \mu_d$), the scattering angle $\theta_{\mathbf{p}+\mathbf{q},\mathbf{p}}$ could be estimated as $\theta_{\mathbf{p}+\mathbf{q},\mathbf{p}} \sim q/p \ll 1$, so that for the scattering rate, one arrives at

$$\frac{1}{\tau_{\text{e} - \text{e}}} \sim \frac{g^2 n_v}{\mu_d^2 / v_F} \left(\frac{T}{A}\right)^{3/\alpha}. \quad (8)$$

Eq. (8) is the main finding of the present work (for the more detailed evaluation, please see Supplementary Information, section I). Namely, this VHS-mediated electron-electron scattering leads to sublinear scaling of transport coefficients.

Let us emphasize that the obtained result is valid only in the limit of Van Hove singularities being sufficiently close to the Fermi level ($T \gg \mu_v$). At this point, it is worth noting that it is challenging to verify whether this condition is truly met. From an experimental perspective, a highly accurate (precision not worse than 10 meV) ARPES measurement is needed. Theoretical DFT predictions may be inaccurate due to ambiguous doping or strain that would shift the Fermi level in a way specific to the experimental setup in question. Nonetheless, in what follows, we demonstrate that the applicability of our model treatment to real systems could be tested in various thermoelectric measurements.

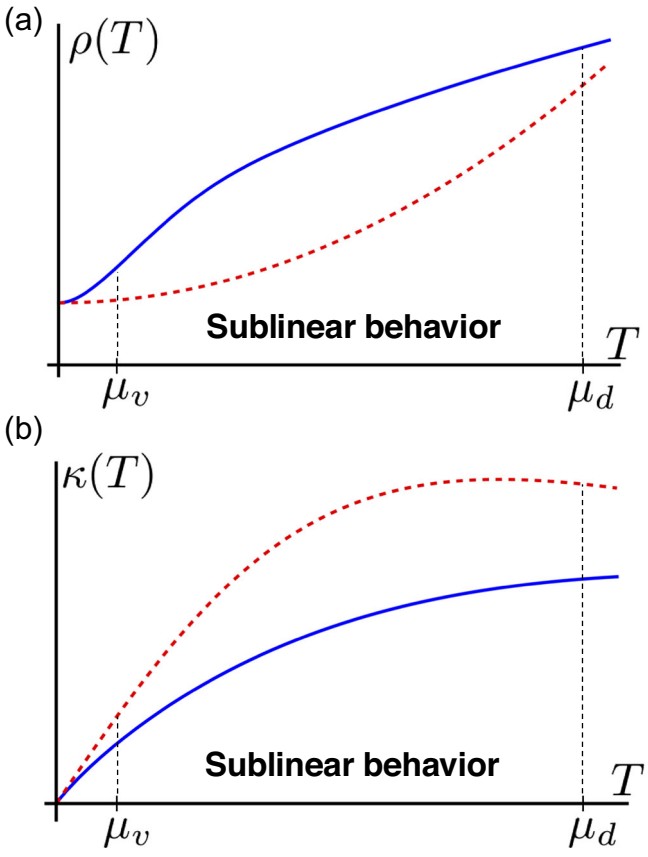

**Fig. 3 | Predicted transport coefficients sublinear scaling with temperature. a** $T$-dependent resistivity $\rho(T)$ for the two-pocket model of a Kagome metal (solid blue line) and a standard Fermi liquid behavior (dashed red line). Kagome metals' resistivity demonstrates sublinear temperature behavior $\rho(T) \propto T^{3/\alpha}$ in a range $\mu_v \ll T \ll \mu_d$ due to the dominant internode electron-electron scattering rate given by Eq. (8). For large $T \gg \mu_d$, intranode interaction becomes the strongest due to lack of screening and thus $\rho(T) \propto T$. **b** $T$-dependent thermal conductivity $\kappa(T)$. Since for $T \ll \mu_d$ Wiedemann-Franz law holds, $\kappa(T)$ also has a sublinear behavior as $\kappa(T) \propto T^{1-3/\alpha}$ in the range $\mu_v \ll T \ll \mu_d$. For $T \gg \mu_d$, however, Wiedemann-Franz law breaks down due to the separation of electric and thermal current relaxation channels (see discussion below).

## Momentum-relaxing processes and sublinear $T$ behavior

One should, however, be aware that momentum-conserving electron-electron interactions alone would lead to non-decaying current; thus, a minimal transport estimation of Kagome metals should include momentum-relaxing processes. As experimental estimates for $ScV_6Sn_6$[24] show, impurity scattering exhibits the shortest scattering time among momentum-relaxing processes. Electron-phonon interactions prove to be negligible[24], since sublinear $T$ behavior is observed below the Debye temperature. Thus, the minimal relaxation term should include internode electron-electron scattering and impurity relaxation.

Within this minimal model of scattering, one could calculate both electrical $\sigma$ and thermal $\kappa$ conductivities for $T \ll \mu_d$ by solving the Boltzmann equation (4):

$$
\begin{aligned}
\sigma &= e^2 \nu_d(\mu_d) D, \quad \kappa = \frac{\pi^2}{3} \nu_d(\mu_d) D T, \\
D &= \frac{1}{2} v_F^2 \tau_{\text{eff}}(\mu_d, T), \quad \tau_{\text{eff}} = \left( \tau_{\text{imp}}^{-1} + \tau_e^{-1} - e \right)^{-1}.
\end{aligned}
\tag{9}
$$

We plot the resistivity and thermal conductivity in Fig. 3. Despite demonstrating non-Fermi liquid-like behavior of sublinear $T$-dependence, Eq. (9) suggests that the Wiedemann-Franz law still holds. This is because both charge and heat currents decay due to the same

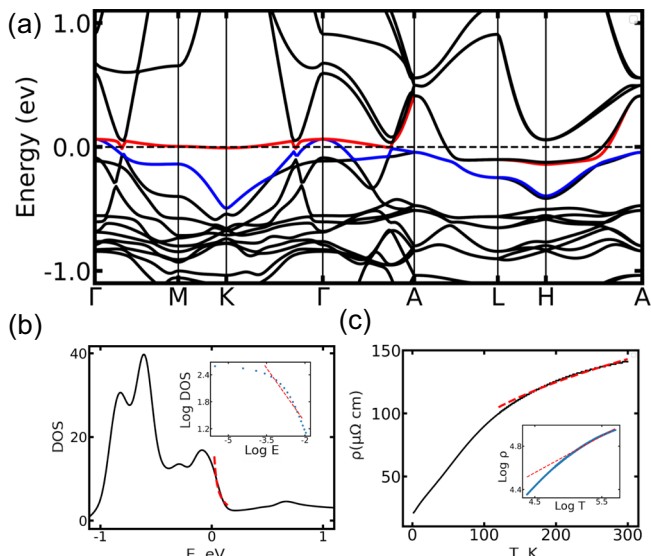

**Fig. 4 | First-principles and experimental resistivity scaling with temperature for Ni$_3$In. a** DFT electronic structure of Ni$_3$In. Fermi level is shown by a dashed line. Red and blue colors represent the two bands closest to the Fermi level. **b** DFT total density of states shows a peak near Fermi level. Dashed red line stands for the fitting as $(E - E_{\text{VHS}})^{-\beta}$, $\beta = 0.8$ (please see inset for log log plot), $E_{\text{VHS}} = 0$. **c** Ni$_3$In experimental resistivity $\rho(T)$[23] versus temperature is shown. Sublinear fitting of Ni$_3$In resistivity $\rho(T) \propto T^\gamma$ gives $\gamma_{\text{exp}} = 0.34$ (red dashed line and inset log-log plot) in accord with our theoretical prediction.

process, namely, internode electron-electron scattering. However, as we discuss later, for high temperatures $T \gg \mu_d$ it is not so, and the Wiedemann-Franz law breaks down.

It is worth mentioning that our predictions could be tested for both thermal conductivity and thermopower measurement. The thermopower, due to Mott relation, is expected under certain conditions (band structure and impurity scattering specific) to demonstrate the same sublinear scaling with thermal conductivity $\propto T^{1-3/\alpha}$. According to recent experimental studies, the predicted sublinear behavior was realized in CsV$_3$Sb$_5$ compound[46]. For ScV$_6$Sn$_6$[47], however, other contributions were dominant, providing for linear behavior.

## Comparison with experimental data

As mentioned in the "Introduction," the well-pronounced sublinear in temperature resistivity was observed[23] in Ni$_3$In compound for $T \gtrsim 100$ K. In the subsequent sections, we illustrate how our two-pocket semiclassical model offers a plausible explanation for this phenomenon.

Band structure calculation for Ni$_3$In (see Fig. 4a) identifies the fast Dirac pocket near $\Gamma$ point. Further, the DFT density of states (see Fig. 4b) demonstrates a peak around Fermi energy. Density of states power law fitting DOS$(E) \propto (E - E_{\text{VHS}})^{-\beta}$ leads to $\beta = 0.8$, a power-law divergent higher order Van Hove singularity[48–50]. According to Eqs. (3) and (8), this predicts resistivity exponent $\gamma_{\text{th}} = \frac{3}{2}(1 - \beta) = 0.3$ ($\rho(T) \propto T^\gamma$). This agrees well with the experimental data fitting result $\gamma_{\text{exp}} = 0.34$. We note the suggested mechanism does not describe the resistivity curve in Fig. 4c below -100 K. As discussed in ref. 23, below 100 K, the resistivity is dominated by strongly non-Fermi liquid correlations. However, for higher temperatures, this correlated state is claimed to be destroyed, falling into the semiclassical regime.

We further perform DFT calculations for other Kagome compounds such as ScV$_6$Sn$_6$, CsV$_3$Sb$_5$, RbV$_3$Sb$_5$, KV$_3$Sb$_5$. In Supplementary Information, section II, we prove with the help of $\mathbf{k} \cdot \mathbf{p}$ model fitting that experimentally observed[24] universal sublinear resistivity behavior agrees well with our theory prediction.

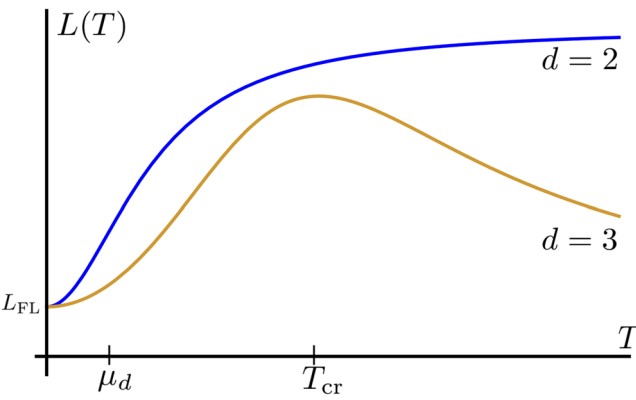

**Fig. 5 | Lorentz number $L(T)$ behavior with temperature $T$.** For $T \ll \mu_d$, Wiedemann-Franz law holds because both charge and heat current relax due to almost elastic internode e-e scattering. For $T \gg \mu_d$, it breaks down since although internode e-e interactions are the strongest, they do not provide thermal current relaxation for $T \gg \mu_d$. For $\mu_d < T < T_{cr}$, thermal current relaxation is provided by internode e-e scattering, which gives nontrivial power-law behavior $L(T) \propto T^\delta$, $\delta = 3 - 3/\alpha$. For higher temperatures $T > T_{cr}$, the increase of $L(T)$ would be cut due to energy-dependent for $T \gg \mu_d$ impurity scattering (the higher temperature behavior also depends on the dimension). $L_{FL} = \frac{\pi^2}{3}\left(\frac{k_B}{e}\right)^2$ is Fermi liquid value of Lorentz number. $T_{cr}$ is a crossover temperature between $\tau_{inter}$ and $\tau_{imp}$ dominated heat current relaxation.

## Wiedemann-Franz law breakdown

Another interesting aspect of the two-pocket model is Wiedemann-Franz law violation at high temperatures case ($T \gtrsim \mu_d$). The electron-hole symmetry is approximately restored within this limit for Dirac node, which essentially implies the absence of screening for intranode interactions (screening length[51] $l_{scr}^2 \propto 1/\nu(\mu_d) \to \infty$). This, in turn, makes the quantitative theoretical description of transport phenomena cumbersome. Hence, the aim of this section is not to give accurate derivations but rather reasonable estimates for the behavior of Lorentz number $L(T)$ with temperature. For $L(T)$, one could write

$$\frac{L(T)}{L_{FL}} \sim \frac{\tau_\kappa}{\tau_{intra}}, \tag{10}$$

where $\tau_\kappa = \min\{\tau_{inter}, \tau_{imp}\}$ describes the thermal current relaxation time. According to ref. 52, intranode electron-electron scattering is unable to relax thermal current (see also Fig. 1c). It is therefore done by internode and impurity scattering. Charge current relaxation time is still given by the shortest time scale (which is now $\tau_{intra}$ due to the absence of screening under approximate electron-hole symmetry). The separation of thermal and charge currents relaxation channels is the ultimate reason for Wiedemann-Franz law breakdown in this limit. Our mechanism is thus in a sense different from previously discussed Wiedemann-Franz law violation in cuprates due to spin-charge separation[53-55]. For the schematic plot of $L(T)$ behavior, please see Fig. 5.

## Discussion

In this work, we have proposed a semiclassical theory to explain sublinear scaling of charge and heat transport coefficients, based on a two-pocket model (Dirac node + Van Hove singularity) of Kagome metals. When Fermi level is sufficiently close to VHS, the dominant scattering source of current-carrying Dirac electrons is internode electron-electron scattering. We have demonstrated that electronic states near high-order Van Hove singularity effectively form a momentum-relaxing reservoir for fast Dirac electrons, allowing only for a vanishing energy exchange. The reservoir size is though $T$-dependent: $q_{VHS} \lesssim (T/A)^{1/\alpha}$. Combined with a relatively large Dirac pocket ($q_D \gg q_{VHS}$) or generic fast electron pocket, this allows for a sublinear $T$-

dependence of transport coefficients, $\rho(T) \propto T^{3/\alpha}$, $\kappa \propto T^{1-3/\alpha}$. Our finding for resistivity scaling directly explains the recent transport experiments in Kagome metal[23,24]. Nevertheless, despite the sublinear behavior at relatively low temperatures, Wiedemann-Franz law still holds since both charge and heat currents relax due to the same internode scattering process. While for relatively large temperatures, the thermal and electric current relaxation channels split due to the approximate electron-hole symmetry of Dirac node[52], leading to Wiedemann-Franz law violation.

## Methods

We performed density functional theory (DFT) calculations using the Full-Potential Local-Orbital (FPLO) code, with the generalized gradient approximation (GGA) in the Perdew-Burke-Ernzerhof (PBE) formulation[56,57]. For the numerical evaluation of the density of states, we employed a dense $250 \times 250 \times 250$ k-point mesh in the Brillouin zone.

## Data availability

The data that support the findings of this study are available from the corresponding author upon request.

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

## Acknowledgements

We are grateful to Hiroki Isobe and Liang Fu for helpful discussions, Shirin Mozaffari and David Mandrus for related experimental collabora-tions. N.M. acknowledges the financial support from the Alexander von

Humboldt Foundation. Y.Z. was supported by the Max Planck partner lab on quantum materials.

## Author contributions

Y.Z. conceived the project. N.P. carried out the charge and heat current calculations, and N.M. performed the DFT calculations. N.P. and Y.Z. wrote the manuscript with input from N.M. and C.F.

## Competing interests

The authors declare no competing interests.
