## [Transparent Peer Review file · Nature Communications]

Sublinear in temperature transport in Kagome metals: Interplay of Dirac cones and Van Hove singularities

Corresponding Author: Professor Yang Zhang

Version 0:

Reviewer comments:

Reviewer #1

(Remarks to the Author)

The authors develop a two-pocket (Dirac + Van Hove) model that explains the sublinear temperature dependence of electrical resistivity observed in numerous kagome metals. By emphasizing how the high density of states near a (possibly higher-order) Van Hove singularity (VHS) amplifies internode electron–electron scattering for Dirac carriers, the model reproduces non-Fermi-liquid-like scaling $\rho(T) \propto T^\gamma$ ($\gamma < 1$). The work also predicts sublinear thermal conductivity and a potential violation of the Wiedemann–Franz law in higher-temperature regimes. Comparisons to Ni_3In and other representative kagome compounds suggest reasonable qualitative agreement.

The present manuscript contains a detailed semiclassical theory of electronic transport, relying on Boltzmann equation approaches and electron–electron scattering near a Van Hove singularity. Such in-depth transport modeling is typically beyond the scope of more general journals such as Nature Communications. I recommend submitting this work to a specialized condensed matter physics journal such as Physical Review B, where the community of readers and referees is well-attuned to advanced transport theory and intricate scattering mechanisms.

Furthermore, the following points are not entirely clear or sufficiently addressed in the manuscript:

- (1) Besides simply listing the VHS position (in meV) in Table I, consider adding more quantitative experimental parameters—e.g., the higher-order VHS exponent β , ARPES-derived Fermi-level shift, effective mass measurements, Hall carrier densities, or residual-resistivity ratios. This expanded data would bolster the analysis of how sublinear exponents relate to the actual DOS enhancement near the VHS.
- (2) There is an inevitable mismatch between DFT-predicted VHS energies and those observed experimentally, partly due to sample-dependent Fermi-level shifts, defects, or strain. Emphasize that the trend—“closer VHS \rightarrow stronger sublinear scaling”—remains robust, but refine the discussion to acknowledge how real samples may differ from ideal DFT values by tens of meV.
- (3) Kagome metals can host multiple saddle points and nontrivial k_z dispersions. If possible, add a short remark (or figure) indicating how multiple VHSs might overlap, or how a higher-order VHS might create a sharper DOS divergence, leading to stronger-than-expected sublinear scaling even if nominally 50–80 meV away from the Fermi level.
- (4) Suggest further thermal-transport (Lorentz ratio) measurements, thermoelectric studies, doping studies, or ARPES experiments pinpointing the VHS location relative to E_F . Such data could verify key assumptions of the model—especially regarding the interplay of intranode versus internode scattering mechanisms.

Overall, the paper presents a compelling, semiclassical explanation for sublinear resistivity in kagome systems, offering valuable insight into electron–electron scattering enhanced by a Van Hove singularity. Once the above points are addressed, I recommend publication in a focused condensed matter journal such as Physical Review B, where the rigorous theoretical and transport-focused content would be best appreciated.

Reviewer #2

(Remarks to the Author)

The manuscript by Peshcherenko et al. presents a concise theoretical model aimed at explaining the sublinear resistivity behavior observed in a broad class of kagome metals. Given the simplicity of the model, its agreement with experimental data is reasonable. The study is timely and relevant to the quantum materials community investigating this emerging class of compounds. However, I find several points of confusion and without the clarification of which the manuscript cannot be accepted.

1. The authors should explicitly define what they mean by "sublinear behavior." I assume they are referring to a resistivity exponent < 1 , but this should be clearly stated at the beginning of the manuscript to avoid ambiguity.
2. The authors cite references [6] and [7] as examples of sublinear temperature dependence in materials near quantum phase transitions. However, both references discuss cases with superlinear resistivity exponents (i.e., exponent > 1). This misrepresentation is particularly misleading to the readers and should be corrected.
3. One of the main concerns in the manuscript is the simplification of the Fermi pocket near the van Hove singularity into an isotropic pocket, as depicted in Fig. 1b. It is well established that Fermi surfaces near van Hove singularities differ significantly from a small, isotropic circular pocket, whether on the electron or hole side. In fact, such Fermi surfaces tend to be considerably larger as one approaches the van Hove singularity. Since the manuscript focuses on Fermi energies near this singularity, and given its crucial role in the scattering process, the validity of this approximation is unclear. The authors must provide proper justification for this simplification and clarify its impact on their conclusions.
4. None of the schematics in Fig. 1 have labeled axes. Proper axis labels are essential for clarity and should be included.
5. The authors should explain why they only consider cases where $\alpha > 2$ in Eq. (2). This restriction should be explicitly justified.
6. Several quantities in the equations are not properly defined. For instance, near Eq. (5), the parameter g is introduced without definition. The authors should ensure that all symbols and parameters are clearly defined when first introduced.

Reviewer #3

(Remarks to the Author)

The authors analyzed the temperature dependence of electric resistivity, electrical conductivity, and thermal conductivity in a system with two Fermi pockets: a Van Hove singularity and a Dirac pocket. As a result, they found that in the intermediate temperature region, the electrical resistivity exhibits a behavior proportional to T^l with $l < 1$, deviating from the conventional Fermi liquid expectation. Furthermore, they demonstrated that the Wiedemann-Franz law is violated in the high-temperature region. The results are simple, and the manuscript is clearly written, but there are some unclear points. Before recommending publication, I suggest revising these points and resubmitting the manuscript.

- The central result of this study is the contribution of internode scattering (Eq. 8) in the region $\mu\nu < T < \mu d$. Does this fully explain all the experimental values of γ in Table 1? (In particular, what causes the differences among the groups with small $\gamma = 0.3, 0.6$ vs large $\gamma = 0.8, 1$)
- Shouldn't the claim that Umklapp scattering is negligible be justified using equations? (Line 153 on page 3.) For example, in Kagome systems, renormalization group analyses of Umklapp scattering have been conducted. [T. Park et al., Phys. Rev. B 104, 035142 (2021)]
- In some Kagome materials, it is known that there are relatively strong long-range Coulomb repulsion. How valid is the present analysis under such conditions?
- Related to the previous question, in these materials, even within a single vHS or Dirac band, orbitals and sublattices may be mixed. Is the present result still applicable in such cases?
- What is the relationship between the position of the vHS and Dirac bands in the Brillouin zone and the obtained results?
- In the present estimation, the scattering matrix at order g^2 is obtained as a perturbation of g in Eq. (5). Up to what magnitude of g is the observed T -sublinear transport allowed? (What is the corresponding effective relaxation time?)
- In Fig. 3(b), would it be possible to plot reference lines similar to those in Fig. 3(a)? Additionally, plotting the violation of the Wiedemann-Franz (WF) law somewhere would make it clearer.
- What is the relationship between the values of a_i in Table 1 of the Supplementary Material and the present results? In the present results, when attempting to fit the vHS band, it appears that even a slight inclusion of the highest-order term in k leads to a continuous decrease in the power of T . Wouldn't this imply that most materials with a vHS would exhibit T -sublinear behavior?

Version 1:

Reviewer comments:

Reviewer #2

(Remarks to the Author)

The revised manuscript by Peshcherenko et al. addresses some of the concerns raised in the previous review. However, significant issues remain, including several unconventional uses of standard terminology. In its current form, the manuscript is not suitable for publication.

1. In the previous review, I questioned the validity of approximating large Fermi surfaces—spanning a substantial fraction of the Brillouin zone near van Hove singularities (see the green contours in the authors' response)—as small, point-like features, as shown in Fig. 1b. After re-reading the text, it appears that what authors are referring to as “Fermi surface size” is not Fermi surfaces themselves, but the population of thermally activated electrons near it (see magenta highlight in attached pdf). This is a nonstandard usage that diverges from textbook definitions and is highly likely to confuse readers.

2. Even under this interpretation is $AqT^\alpha \sim T$ still true?

3. Related to the above, the authors should clarify which electrons in the magenta region are being modeled. It is not clear whether the theory intended to capture all the electrons on the triangular Fermi surfaces, or just the portion near M point.

4. In the Introduction, the authors should clarify that non-Fermi liquid behavior encompasses both linear-in-TTT (“strange metal”) and super-linear temperature dependence.

5. For exponents summarized in Table I, the authors should specify the temperature ranges over which the power-law fits were performed.

6. The expression on line 138 “provide nice... physics” is subjective and should be revised for scientific objectivity.

7. The authors tend to overuse the term “non-Fermi liquid” throughout the manuscript. While it can literally refer to any state deviating from a Fermi liquid behavior, the community conventionally reserves this term for low temperature deviations from $\alpha=2$ behavior. The use of “non-Fermi liquid” to describe high-temperature deviations is misleading and should be reconsidered. Line 182 is one example, but this issue occurs in multiple places.

8. In the revised manuscript, the authors predict a power law evolution of thermopower from their transport theory. Relevant experimental data on ScV₆Sn₆ (see e.g. Applied Phys. Lett. 125, 152202) and CsV₃Sb₅ (Phys. Rev. B 104, L180508) are already available and should be discussed in comparison with the predictions.

Reviewer #3

(Remarks to the Author)

The authors appear to have generally addressed the reviewers' questions, and the manuscript seems to have been improved.

Indeed, the content of the manuscript is somewhat technical, but overall, it appears to meet the publication standards of Nature Communications.

Therefore, I recommend it for publication.

Version 2:

Reviewer comments:

Reviewer #2

(Remarks to the Author)

The authors have addressed most of my concerns and the clarity of the manuscript has improved. In the current manuscript there are still descriptions (say in Fig. 1a caption) that van Hove singularity hosts a small Fermi surface, which can cause confusion because with van Hove singularity the Fermi surface of the system is typically large.

REVIEWER COMMENTS

Reviewer #1 (Remarks to the Author):

The authors develop a two-pocket (Dirac + Van Hove) model that explains the sublinear temperature dependence of electrical resistivity observed in numerous kagome metals. By emphasizing how the high density of states near a (possibly higher-order) Van Hove singularity (VHS) amplifies internode electron–electron scattering for Dirac carriers, the model reproduces non-Fermi-liquid-like scaling $\rho(T) \propto T^\gamma$ ($\gamma < 1$). The work also predicts sublinear thermal conductivity and a potential violation of the Wiedemann–Franz law in higher-temperature regimes. Comparisons to Ni₃In and other representative kagome compounds suggest reasonable qualitative agreement.

The present manuscript contains a detailed semiclassical theory of electronic transport, relying on Boltzmann equation approaches and electron–electron scattering near a Van Hove singularity. Such in-depth transport modeling is typically beyond the scope of more general journals such as Nature Communications. I recommend submitting this work to a specialized condensed matter physics journal such as Physical Review B, where the community of readers and referees is well-attuned to advanced transport theory and intricate scattering mechanisms.

Reply: We sincerely thank the reviewer for taking time to evaluate our manuscript, and the expert summary of our work. However, we respectfully disagree that our study is overly specialized. On the contrary, our work aims to address anomalous transport behaviors that are broadly observed across a wide class of kagome materials as summarized in Table I, and provides a general theoretical framework that can be of interest to a wider condensed matter audience.

Regarding the scope and suitability of the work, we would like to respectfully clarify that while our formalism employs a semiclassical Boltzmann approach, the core conceptual advance lies in how a minimal and physically transparent two-carrier (Dirac + VHS) model naturally explains the widespread anomalous transport behavior in Kagome materials. This includes non Fermi Liquid behavior such as the sublinear-in-temperature resistivity scaling and breakdown of the Wiedemann–Franz law—phenomena that have been observed experimentally but lack a broadly accepted microscopic explanation.

Importantly, our model does not rely on fine-tuning or material-specific parameters, but rather highlights a general mechanism based on the coexistence of Dirac bands and Van Hove singularities—a feature ubiquitous in many kagome systems and beyond. We believe this universality, combined with the simplicity of the framework, makes the work accessible and relevant to a broad condensed matter audience, including experimentalists who seek guidance in interpreting puzzling transport signatures.

We believe that the conceptual clarity and broad applicability of our model justify consideration by a wider readership interested in emergent quantum phenomena in quantum materials. We hope the revised manuscript more clearly communicates this perspective.

Furthermore, the following points are not entirely clear or sufficiently addressed in the manuscript:

(1) Besides simply listing the VHS position (in meV) in Table I, consider adding more quantitative experimental parameters—e.g., the higher-order VHS exponent β , ARPES-derived Fermi-level shift, effective mass measurements, Hall carrier densities, or residual-resistivity ratios. This expanded data would bolster the analysis of how sublinear exponents relate to the actual DOS enhancement near the VHS.

Reply: We thank the referee for this great suggestion. For our proposal, the crucial features of Kagome metals are Van Hove singularities (hosting slow electrons with large masses) and a fast pocket (providing fast electrons with low masses). To prove these features to be present in real materials, we have expanded the table I in the revised version of the manuscript. Namely, we have included ARPES evidence for Van Hove singularities and listed relatively low effective electron masses (extracted from de Haas-van Alfen oscillations) to signify the presence of a fast Dirac pocket. We hope that the presented extra experimental data could provide more evidence why the model considered in this study is reasonable.

(2) There is an inevitable mismatch between DFT-predicted VHS energies and those observed experimentally, partly due to sample-dependent Fermi-level shifts, defects, or strain. Emphasize that the trend—“closer VHS \rightarrow stronger sublinear scaling”—remains robust, but refine the discussion to acknowledge how real samples may differ from ideal DFT values by tens of meV.

Reply: We sincerely appreciate the reviewer’s insightful suggestion. While DFT-calculated energies may indeed deviate from experimental values, we have supplemented our analysis with additional experimental data (see table I) in the revised manuscript to confirm key band structure features. As recommended, we have expanded the discussion in the “Internode scattering” section to further contextualize these findings. Importantly, our proposed model should be interpreted as *one possible* minimal semiclassical framework for the observed sublinear scaling behavior.

(3) Kagome metals can host multiple saddle points and nontrivial k_z dispersions. If possible, add a short remark (or figure) indicating how multiple VHSs might overlap, or how a higher-order VHS might create a sharper DOS divergence, leading to stronger-than-expected sublinear scaling even if nominally 50–80 meV away from the Fermi level.

Reply: We thank the reviewer for raising this excellent point. In our work we focus on the case of Ni_3In that hosts a non-degenerate band around M point (see Fig. 1 below), making the degenerate VHS discussion irrelevant. However, other examples of kagome materials (e.g.,

AV₃Sb₅ materials discussed in supplementary information) definitely allow for degenerate Van Hove singularities (see Fig. 2 below). In this case, we expect that the contributions of every VHS to the scattering rate is dominant in the limit $\tau \rightarrow \infty$, otherwise no sublinear contribution is expected. From Fig. 2 it is straightforward to see that the distance between two Van Hove singularities positions is relatively large in all cases (0.035 eV for ScV₆Sn₆, 0.1 eV for CV₃Sb₅, 0.04 eV for RbV₃Sb₅ and KV₃Sb₅), so that the second VHS is likely to contribute only for temperatures approaching 400 K, which is the upper end of sublinear domain in present experiments.

Fig. 1 Non-degenerate VHS in Ni₃In around M point.

Fig. 2 VHSs overlap around M point for (a) ScV₆Sn₆, (b) CV₃Sb₅, (c) RbV₃Sb₅, (d) KV₃Sb₅.

(4) Suggest further thermal-transport (Lorentz ratio) measurements, thermoelectric studies, doping studies, or ARPES experiments pinpointing the VHS location relative to E_F. Such data could verify key assumptions of the model—especially regarding the interplay of intranode versus internode scattering mechanisms.

Reply: We thank the reviewer for this excellent remark. Beyond already suggested in the main text electrical and thermal conductivity measurements, one might want to consider a thermopower measurement. The thermopower, due to Mott relation, is expected to demonstrate the same sublinear scaling with thermal conductivity ($T^{1-3/\alpha}$). A perfect platform for testing this prediction could be Vanadium-based kagome materials. These kagome metals have been proved [S. Mozaffari et al, Phys Rev B **110**, 035135 (2024)] to exhibit a CDW transition, which effectively shifts Fermi level closer to VHS position. Thus, the expected result for thermopower is sublinear scaling above and trivial T linear behavior below the transition.

Another possible direction of further experimental verification is to shift the Fermi level position by doping. The possibility of doping away the Fermi level from the VHS was previously demonstrated for CsTi₃Be₅ in [Bo Liu, Physical Review Letters **131**, 026701 (2023)] with no structural instability involved. Performing identical sets of thermoelectric measurements for doped and undoped samples would therefore help to test our theory predictions. Since our theory hinges on the Van Hove singularity being close to the Fermi level, for Fermi level being away from the VHS the prediction to test would be to have sublinear scaling only for one sample but not for the both.

Overall, the paper presents a compelling, semiclassical explanation for sublinear resistivity in kagome systems, offering valuable insight into electron–electron scattering enhanced by a Van Hove singularity. Once the above points are addressed, I recommend publication in a focused condensed matter journal such as Physical Review B, where the rigorous theoretical and transport-focused content would be best appreciated.

Reply: We thank the reviewer for the expert summary of the scientific results and for recognizing the novelty of our approach. In the revised manuscript, we highlight that our work builds on a simple semiclassical Boltzmann framework involving two types of carriers—Dirac electrons and electrons near a Van Hove singularity—whose coexistence is a generic feature in many kagome metals. Despite the simplicity of the model, it captures a range of striking transport anomalies, including sublinear resistivity, suppressed thermal conductivity, and potential violation of the Wiedemann–Franz law.

The strength of our approach lies in its physical transparency and broad applicability. Without invoking exotic interactions or material-specific tuning, it provides a natural explanation for non-Fermi-liquid-like behavior observed across multiple kagome systems. We believe this insight will be valuable not only to transport theorists, but also to experimentalists seeking a unifying framework to interpret anomalous transport signatures in a broad class of correlated materials.

Reviewer #2 (Remarks to the Author):

The manuscript by Peshcherenko et al. presents a concise theoretical model aimed at explaining the sublinear resistivity behavior observed in a broad class of kagome metals. Given the simplicity of the model, its agreement with experimental data is reasonable. The study is timely and relevant to the quantum materials community investigating this emerging class of compounds. However, I find several points of confusion and without the clarification of which the manuscript cannot be accepted.

Reply: We sincerely thank the reviewer for the positive assessment of our work and for recognizing its relevance and timeliness to the quantum materials community. We also appreciate the constructive feedback and the identification of points requiring clarification. We take these comments seriously and have carefully revised the manuscript to address all concerns in detail. We believe the revised version resolves the ambiguities and presents our results with greater clarity.

1. The authors should explicitly define what they mean by "sublinear behavior." I assume they are referring to a resistivity exponent < 1 , but this should be clearly stated at the beginning of the manuscript to avoid ambiguity.

Reply: We thank the reviewer for bringing this point. By sublinear behavior we indeed meant the resistivity exponent γ from the relation $\rho(T) \propto T^\gamma$ being $\gamma < 1$. We have fully implemented the reviewer's suggestion and amended the beginning of the manuscript.

2. The authors cite references [6] and [7] as examples of sublinear temperature dependence in materials near quantum phase transitions. However, both references discuss cases with superlinear resistivity exponents (i.e., exponent > 1). This misrepresentation is particularly misleading to the readers and should be corrected.

Reply: We thank the reviewer for bringing our attention to this issue, this typo is now fixed. Initially we purported to provide only non-Fermi liquid behavior examples and not sublinear scaling.

3. One of the main concerns in the manuscript is the simplification of the Fermi pocket near the van Hove singularity into an isotropic pocket, as depicted in Fig. 1b. It is well established that Fermi surfaces near van Hove singularities differ significantly from a small, isotropic circular pocket, whether on the electron or hole side. In fact, such Fermi surfaces tend to be considerably larger as one approaches the van Hove singularity. Since the manuscript focuses on Fermi energies near this singularity, and given its crucial role in the scattering process, the validity of this approximation is unclear. The authors must provide proper justification for this simplification and clarify its impact on their conclusions.

Reply: We thank the reviewer for raising this excellent point. Indeed, Fermi pockets near the M point are likely anisotropic (see Fig. 3 below). Nevertheless, we expect our sublinear behavior prediction to hold true.

The reason is that the proposed scattering mechanism is a direct generalization of Fermi liquid T^2 behavior, in a sense that both T^2 and sublinear T^γ behaviors arise from the phase volume of scattered electrons. The difference in the end result stems from the Van Hove singularity (with band dispersion $E \sim Aq^\alpha$) Fermi surface size q_T being sub linearly dependent on T: $q_T \propto (T/A)^{1/\alpha}$. The latter is not the case for Fermi liquid.

Thus, we expect the Fermi surface size estimate $q_T \propto (T/A)^{1/\alpha}$ to hold true even for non-isotropic Fermi surfaces. So that, even in the non-isotropic case the corresponding phase volume would scale sublinearly, so that our primary findings (sublinear scaling of resistivity with T) would still hold true.

Fig. 3 Fermi surface of a 6 band kagome metal (model calculation).

4. None of the schematics in Fig. 1 have labeled axes. Proper axis labels are essential for clarity and should be included.

Reply: We thank the reviewer for the suggestion. Following the reviewer's comment, we have fixed the axes labeling in the revised version as below:

5. The authors should explain why they only consider cases where $\alpha > 2$ in Eq. (2). This restriction should be explicitly justified.

Reply: We thank the reviewer for this comment. The reason is that for $\alpha \leq 2$ in a 2D system the density of states exhibits no Van Hove singularity. No Van Hove singularity would make interpocket exchange (which is crucial for the sublinearity) of the same or lesser order of magnitude with superlinear intrapocket one. This would in turn render sublinear features non-observable. To improve the presentation towards readability, we have included the corresponding remark in the revised version.

6. Several quantities in the equations are not properly defined. For instance, near Eq. (5), the parameter g is introduced without definition. The authors should ensure that all symbols and parameters are clearly defined when first introduced.

Reply: We are extremely grateful to the reviewer for pointing this out. We have introduced the definition of interaction amplitude g near Eq. (5) and have thoroughly polished the manuscript to avoid further inconsistencies.

Reviewer #3 (Remarks to the Author):

The authors analyzed the temperature dependence of electric resistivity, electrical conductivity, and thermal conductivity in a system with two Fermi pockets: a Van Hove singularity and a Dirac pocket. As a result, they found that in the intermediate temperature region, the electrical resistivity exhibits a behavior proportional to T^l with $l < 1$, deviating

from the conventional Fermi liquid expectation. Furthermore, they demonstrated that the Wiedemann-Franz law is violated in the high-temperature region. The results are simple, and the manuscript is clearly written, but there are some unclear points. Before recommending publication, I suggest revising these points and resubmitting the manuscript.

Reply: We sincerely thank the reviewer for the positive assessment of our work and for highlighting the clarity and simplicity of our approach. We appreciate the careful reading and the constructive suggestions. Below, we respond in detail to each of the points raised, and have revised the manuscript accordingly to address all concerns.

•The central result of this study is the contribution of internode scattering (Eq. 8) in the region $\mu\nu < T < \mu d$. Does this fully explain all the experimental values of γ in Table 1? (In particular, what causes the differences among the groups with small $\gamma=0.3, 0.6$ vs large $\gamma=0.8, 1$)

Reply: The reason for different γ 's is simply the different band bending for Van Hove singularity around M point. The trend here is the following: the higher order Van Hove singularity, the lesser the sublinearity exponent. For Ni_3In , the band around M point is almost flat, making it for $\gamma = 0.3$. For V-based kagome compounds, as shown in the Supplementary Material, the experimental data fitting suggests $\gamma = 0.6$, corresponding to the 5th order Van Hove singularity ($E(k) \propto |k_x|^5 - |k_y|^5$). As for $\gamma = 1$ case, we view it as a signature of other mechanisms being dominant (such as electron-phonon scattering in diluted metals, [E. Hwang and S. Das Sarma, Phys Rev B 99, 085105 (2019)]), resulting from either VHS being away from Fermi level or the electron-phonon coupling being strong.

In general, the main message of our work is to propose a possible semiclassical mechanism that could explain why some materials exhibit sublinear behavior.

•Shouldn't the claim that Umklapp scattering is negligible be justified using equations? (Line 153 on page 3.) For example, in Kagome systems, renormalization group analyses of Umklapp scattering have been conducted. [T. Park et al., Phys. Rev. B 104, 035142 (2021)]

Reply: We thank the referee for the insightful remark. In [T. Park et al] the authors have demonstrated that including an Umklapp term could lead to the formation of new phases (either superconducting or CDW/SDW). For materials studied in our work, any non-trivial phase is most likely washed out due to temperature fluctuations. For instance, a kagome compound ScV_6Sn_6 is known [S. Mozzafari et al, Phys. Rev. B **110**, 035135 (2024)] to exhibit the CDW phase at low temperatures ($T < 90$ K). At the same time, the non-linear scaling is present for higher temperatures (in the domain from 100 K to 400 K), asserting that the domain of sublinearity is weak Umklapp scattering domain. Hence, we don't expect the Umklapp process to play an essential role.

Beyond that, the sublinear scattering channel is operable only for the transferred momentum being smaller than the Fermi wavevector. Due to the relatively low concentration of electrons in Kagome metals, the transferred momentum doesn't exceed 10^7 cm^{-1} . However, based on the lattice step $a = 5 \text{ \AA}$, the Brillouine zone size (and the corresponding Umklapp momentum transfer) is of the order of $2 \cdot 10^8 \text{ cm}^{-1}$, implying suppressed Umklapp scattering for materials in question.

•In some Kagome materials, it is known that there are relatively strong long-range Coulomb repulsion. How valid is the present analysis under such conditions?

Reply: We thank the reviewer for raising the concern regarding strong long-range Coulomb repulsions. In the case of unscreened Coulomb interaction both intranode and internode scattering rates could be modified. Assuming Fermi level to be away from charge neutrality ($T \ll \mu_d$), the intranode scattering is likely to remain screened (due to relatively small intranode momentum transfer). In the unscreened limit, internode scattering potential takes the following form in Fourier space:

$$V(q) = \frac{2\pi}{|q|},$$

so that the corresponding integral for the transport scattering time (compare with the derivation of Eq. (A9) in the appendix) is given by

$$\frac{1}{\tau_{tr}} \propto \int_{|q| < q_T} d^2 q V^2(q) \delta(E(p) - E(p + q)) (1 - \cos \theta_{p, p+q}) \propto \int_0^{q_T} \frac{dq}{q^2 \sqrt{1 - q^2/4p_F^2}} \left(\frac{q}{2p_F}\right)^2 \propto q_T \propto \left(\frac{T}{A}\right)^{1/\alpha}$$

where the integration is performed over transmitted momentum q . As could be seen from the result above, long-range Coulomb interaction would still lead to sublinear scaling, with sublinearity being even more well-pronounced.

•Related to the previous question, in these materials, even within a single vHS or Dirac band, orbitals and sublattices may be mixed. Is the present result still applicable in such cases?

Reply: We thank the reviewer for raising the concern regarding mixing of orbitals and sublattices. The dominant source of sublinear behavior in our scattering channel arises from the phase space volume of scattered electrons. Consequently, for systems with multiple orbitals, the total scattering rate should be determined by summing contributions from all pairs of degenerate Dirac and van Hove singularity (VHS) bands. In scenarios where orbital mixing induces band splitting, we anticipate that the VHS closest to the Fermi level will dominate the scattering rate.

•What is the relationship between the position of the vHS and Dirac bands in the Brillouin zone and the obtained results?

Reply: We thank the reviewer for the comment on momentum space position. The crucial part of our setup is only to have the VHS close to the Fermi level, so that the contribution of interpocket electron-electron exchange is enhanced. Absolute or relative positions of these two pockets would not change our prediction as soon as the Fermi level is tuned away from the charge neutrality point for Dirac semimetal.

•In the present estimation, the scattering matrix at order g^2 is obtained as a perturbation of g in Eq. (5). Up to what magnitude of g is the observed T-sublinear transport allowed? (What is the corresponding effective relaxation time?)

Reply: We thank the reviewer for bringing the interaction strength. For the contact potential interaction, the non-Born corrections to scattering amplitude are negligible given that $gv_T \ll 1$, where $v_T = \frac{\hbar v_F q_T^3}{\mu_d^2}$ is the effective DOS of current-carrying states (see Eqs. (A9) - (A10) in the appendix). This estimate suggests that for high enough temperatures the lowest order approximation would indeed be broken. However, this temperature scale is likely to exceed Debye temperature, above which the scattering on phonons would diminish sublinear behavior.

•In Fig. 3(b), would it be possible to plot reference lines similar to those in Fig. 3(a)? Additionally, plotting the violation of the Wiedemann-Franz (WF) law somewhere would make it clearer.

Reply: We are very grateful to the reviewer for pointing this out. In the revised version of the manuscript we have put the updated version of Fig. 3 as here:

We have also moved the WF law violation plot from the appendix to the main text as Figure 5:

•What is the relationship between the values of a_i in Table 1 of the Supplementary Material and the present results? In the present results, when attempting to fit the vHS band, it appears that even a slight inclusion of the highest-order term in k leads to a continuous decrease in the power of T . Wouldn't this imply that most materials with a vHS would exhibit T-sublinear behavior?

Reply: We thank the reviewer for this insightful comment on vHS. The results of appendix C suggest that if the density of states demonstrates a high order power-law VHS, then indeed a scattering channel allowing for sublinear scaling exists. Since cubic terms alone could not provide for a power-law density of states singularity, we attribute the singular behavior to k^5 term. It does not mean, however, that including any arbitrary small fifth order term would lead to T sublinear behavior. This is due to electron-phonon coupling constant left unknown in different materials, which can dominate the e-e contribution suggested in our work. For instance, in the material YV_6Sn_6 [Yang et al., <https://arxiv.org/abs/2402.03765>] no sublinear scaling was observed, despite Fermi level being sufficiently close to VHS position, due to the dominance of other superlinear scattering channels. Thus, the central message of our work is that a broadly applicable scattering mechanism capable of producing sublinear resistivity scaling exists—but it need not be the dominant channel in all materials with vHS.

Reviewer #2 (Remarks to the Author):

The revised manuscript by Peshcherenko et al. addresses some of the concerns raised in the previous review. However, significant issues remain, including several unconventional uses of standard terminology. In its current form, the manuscript is not suitable for publication.

We thank the reviewer for taking time to review the updated version of our manuscript and for the thoughtful and constructive feedback provided, which has helped us to improve the manuscript. In the following, we address each of the reviewer's comments point by point.

1. In the previous review, I questioned the validity of approximating large Fermi surfaces—spanning a substantial fraction of the Brillouin zone near van Hove singularities (see the green contours in the authors' response)—as small, point-like features, as shown in Fig. 1b. After re-reading the text, it appears that what authors are referring to as “Fermi surface size” is not Fermi surfaces themselves, but the population of thermally activated electrons near it (see magenta highlight in attached pdf). This is a nonstandard usage that diverges from textbook definitions and is highly likely to confuse readers.

We are grateful to the reviewer for raising this important point. Indeed, the sublinear behavior our theory predicts relies on the size of the relevant momentum space region in the vicinity of VHS being sublinearly T dependent. This is only true for Fermi level being sufficiently close to the VHS (so that $T \ll \mu_v$, where μ_v is the distance between the Fermi level and VHS position).

We agree that our previous use of the term “VHS Fermi surface” was nonstandard and could lead to confusion. As the reviewer correctly notes, we were referring not to the Fermi surface itself, but to the region of thermally activated carriers near the VHS. We have now clarified this distinction explicitly in the revised manuscript to avoid ambiguity and ensure consistency with standard terminology.

2. Even under this interpretation is $AqT^\alpha \sim T$ still true?

We thank the reviewer for this thoughtful follow-up question. We believe it should hold true since the most important electrons in our approach are the ones closest to the VHS point. Thus, the energy width of thermally activated electronic states near the VHS could be estimated as proportional to T . This energy width, in turn, would give rise to temperature sublinear ‘thermally activated’ domain size due to the superlinear nature of band dispersion near the VHS.

3. Related to the above, the authors should clarify which electrons in the magenta region are being modeled. It is not clear whether the theory intended to capture all the electrons on the triangular Fermi surfaces, or just the portion near M point.

We thank the reviewer for raising this question. Our theory specifically targets the electrons in the vicinity of the van Hove singularity at the M point. These electrons dominate the response because the interpocket electron-electron exchange is strongest near the VHS, where the density of states exhibits singular behavior. In contrast, electrons farther away on the triangular Fermi surfaces contribute much less significantly for the scattering, as their associated density of states is nonsingular and their interactions are comparatively weaker.

4. In the Introduction, the authors should clarify that non-Fermi liquid behavior encompasses both linear-in- T ("strange metal") and super-linear temperature dependence.

We are grateful to the reviewer for this helpful suggestion regarding the introduction. In the revised version we have explicitly clarified that non-Fermi liquid behavior can encompass both linear-in- T ("strange metal") and superlinear temperature dependence, to provide a more complete and accurate context for our work.

5. For exponents summarized in Table I, the authors should specify the temperature ranges over which the power-law fits were performed.

We are grateful to the reviewer for the suggestion on the summary table. In the revised version, we have expanded Table I following the reviewer's suggestion. We note that the lower limit of sublinear temperature range in the case of V-based kagome metals corresponds to a CDW transition temperature. This CDW transition, in turn, shifts the VHS close to Fermi level, thus opening the sublinear transport channel. The resistivity measurements are typically performed up to room temperature, but we do expect that the sublinear behaviour persists to even higher temperatures approaching Debye temperature.

6. The expression on line 138 "provide nice... physics" is subjective and should be revised for scientific objectivity.

We thank the reviewer for the comment. In the present version we have replaced 'provides nice... physics' with 'reveals additional observable physical phenomena.'

7. The authors tend to overuse the term "non-Fermi liquid" throughout the manuscript. While it can literally refer to any state deviating from a Fermi liquid behavior, the community conventionally reserves this term for low temperature deviations from $\alpha=2$ behavior. The use of "non-Fermi liquid" to describe

high-temperature deviations is misleading and should be reconsidered. Line 182 is one example, but this issue occurs in multiple places.

We fully agree with the reviewer's suggestion. In order to eliminate a potential source of confusion, we have adapted the notion of 'sublinear behavior' instead of "non-Fermi liquid" and thoroughly polished the manuscript to avoid any future misunderstanding.

8. In the revised manuscript, the authors predict a power law evolution of thermopower from their transport theory. Relevant experimental data on ScV₆Sn₆ (see e.g. Applied Phys. Lett. 125, 152202) and CsV₃Sb₅ (Phys. Rev. B 104, L180508) are already available and should be discussed in comparison with the predictions.

We are extremely grateful to the referee for bringing in this excellent point. To begin with, one should be aware that our prediction for sublinear thermopower scaling is not as robust as the corresponding predictions for resistivity and thermal conductivity. Namely, according to the updated version of the thermopower scaling discussion, the scaling could be either linear in T or sublinear, depending on the details of the material, such as the relative behavior of impurity and e-e scattering as a function of Fermi level. These fine requirements originate from the Mott relation containing derivatives with respect to the chemical potential μ :

$$S(T) = \frac{\pi^2 k_B^2 T}{3e} \frac{\sigma'(\mu)}{\sigma(\mu)}, \quad \sigma(\mu) = e^2 v_d(\mu) \tau_{eff}(\mu).$$

Under certain conditions (such as the presence of multiple electron and hole current-carrying pockets), the temperature dependence of the thermopower can exhibit sublinear scaling $S(T) \propto T^{1-3/\alpha}$, with α being the VHS exponent ($E_{VHS}(q) \propto q^\alpha$). If these conditions remain unfulfilled, the thermopower would behave linear with T.

Despite thermopower scaling prediction being non-universal, it nonetheless could be used to provide an intuitive semiclassical picture of thermopower sublinear scaling. For instance, the CsV₃Sb₅ data (Phys. Rev. B 104, L180508) demonstrates a well-pronounced sublinear fitting starting from 100 K and up to the room temperature. The fitting (see Fig. 1 below) predicts sublinear scaling $S(T) \propto T^n$ with $n_{exp} = 0.43$. Based on CsV₃Sb₅ band structure behavior presented in SI, our theory based on Mott relation predicts scaling with $n_{th} = 0.4$, making for a good agreement with experimental results. The data for ScV₆Sn₆ from Applied Phys. Lett. 125, 152202 however demonstrates linear behavior for T>200 K due to the dominance of other contributions.

In the revised manuscript, we have incorporated a discussion of the two papers suggested by the referee and expanded our analysis of the possible sublinear scaling of thermopower.

Fig. 1 Sublinear scaling of thermopower coefficient for CsV₃Sb₅. The fitting of $S(T)$ is performed with an offset power law function of temperature: $f(x) = a + bx^n$.

Reviewer #3 (Remarks to the Author):

The authors appear to have generally addressed the reviewers' questions, and the manuscript seems to have been improved.

Indeed, the content of the manuscript is somewhat technical, but overall, it appears to meet the publication standards of Nature Communications.

Therefore, I recommend it for publication.

We sincerely thank the reviewer for their time and constructive feedback, along with the positive assessment of our work. We are pleased that the revisions have addressed the referee's concerns and are grateful for their recommendation for publication in Nature Communications.

REVIEWER COMMENTS

Reviewer #2 (Remarks to the Author):

The authors have addressed most of my concerns and the clarify of the manuscript has improved. In the current manuscript there are still descriptions (say in Fig. 1a caption) that van Hove singularity hosts a small Fermi surface, which can cause confusion because with van Hove singularity the Fermi surface of the system is typically large.

We thank the reviewer for this suggestion. We fully agree that the descriptor 'small Fermi surface' was potentially confusing, as the actual Fermi pockets near the VHS can be large. To address this, we have revised the relevant descriptions in both the figures description and the main text to clarify that our analysis focuses on the limited momentum space region immediately surrounding the van Hove singularity point. The term "small Fermi surface" has been removed from the Figures 1,2 captions, with the explicit description of this limited in momentum space region added in the main text on line 109.

We believe this revision eliminates any potential for confusion and thank the reviewer again for suggesting this important clarification.

The revised manuscript by Peshcherenko et al. addresses some of the concerns raised in the previous review. However, significant issues remain, including several unconventional uses of standard terminology. In its current form, the manuscript is not suitable for publication.

1. In the previous review, I questioned the validity of approximating large Fermi surfaces—spanning a substantial fraction of the Brillouin zone near van Hove singularities (see the green contours in the authors’ response)—as small, point-like features, as shown in Fig. 1b. After re-reading the text, it appears that what authors are referring to as “Fermi surface size” is not Fermi surfaces themselves, but the population of thermally activated electrons near it (see magenta highlight). This is a nonstandard usage that diverges from textbook definitions and is highly likely to confuse readers.

2. Even under this interpretation is $AqT^\alpha \sim T$ still true?
3. Related to the above, the authors should clarify which electrons in the magenta region are being modeled. It is not clear whether the theory intended to capture all the electrons on the triangular Fermi surfaces, or just the portion near M point.
4. In the Introduction, the authors should clarify that non-Fermi liquid behavior encompasses both linear-in-TTT ("strange metal") and super-linear temperature dependence.

5. For exponents summarized in Table I, the authors should specify the temperature ranges over which the power-law fits were performed.
6. The expression on line 138 “provide nice... physics” is subjective and should be revised for scientific objectivity.
7. The authors tend to overuse the term “non-Fermi liquid” throughout the manuscript. While it can literally refer to any state deviating from a Fermi liquid behavior, the community conventionally reserves this term for low temperature deviations from $\alpha=2$ behavior. The use of “non-Fermi liquid” to describe high-temperature deviations is misleading and should be reconsidered. Line 182 is one example, but this issue occurs in multiple places.
8. In the revised manuscript, the authors predict a power law evolution of thermopower from their transport theory. Relevant experimental data on ScV₆Sn₆ (see e.g. Applied Phys. Lett. 125, 152202) and CsV₃Sb₅ (Phys. Rev. B 104, L180508) are already available and should be discussed in comparison with the predictions.